# Prevalence and socio-demographic correlates of depression and anxiety among late adolescents (15 to 21 years) in Mymensingh division, Bangladesh: A cross-sectional study

Roni Khatun[1], Salma Akter Urme[2], Md. Syful Islam[3]*

**1** Department of Population Science, Jatiya Kabi Kazi Nazrul Islam University, Trishal, Mymensingh, Bangladesh, **2** Department of Anthropology, Comilla University, Cumilla, Bangladesh, **3** Department of Population Sciences, University of Dhaka, Dhaka, Bangladesh

☯ These authors contributed equally to this work.
* syful111@gmail.com

## Abstract

### Background

Depression and anxiety are prevalent mental health disorders among adolescents worldwide, including Bangladesh. However, mental health disorders are often overlooked in developing countries like Bangladesh. This study aims to assess the prevalence of depression and anxiety, as well as the socio-demographic factors that contribute to these conditions, among late adolescents aged 15–21 in Mymensingh division, Bangladesh.

### Methods

This cross-sectional study was conducted among 400 adolescents aged 15–21 in the Mymensingh division of Bangladesh. The study used a structured questionnaire to collect sociodemographic and lifestyle-related information from the respondents. The questionnaire included PHQ-9 and GAD-7 scales to assess depression and anxiety among the respondents. Descriptive statistics, chi-square tests, and ordinal logistic regression were performed as statistical analyses of this study.

### Results

The overall prevalence of depression was 47.8% (33.5% moderate, 10.5% moderately severe, and 3.8% severe depression), and anxiety 32.8% (31% moderate and 1.8% severe anxiety). Across most of the socio-demographic and lifestyle-related factors, the prevalence of depression and anxiety was higher among adolescent boys. Individuals aged 19–21 are most likely to experience depression and anxiety. The study found that unmarried adolescents, mobile phone users, rural adolescents, and

which permits unrestricted use, distribution, and reproduction in any medium, provided the original author and source are credited.

**Data availability statement:** The data generated and analyzed in this study will be securely archived in institutional archives and appropriately backed up to ensure long-term preservation. The data will be available at reasonable request to the corresponding author, allowing persistent access. This approach ensures that the data can be shared while maintaining the appropriate level of control regarding its distribution and use. Data-sharing requests should be made to (contact via syful111@gmail.com), and access will be provided according to relevant ethical guidelines and institutional policies. Md. Nure Alam Siddiqi, Assistant Professor of the Department of Population Science and the signatory of the Ethical Clearance of this study will be the non-author contact (contact via nurealam_rasel@yahoo.com) for the data access of this study. Md. Nure Alam Siddiqi did not collaborate in the study and is not listed as an author in the manuscript. Besides the corresponding author, Mr. Siddiqi will also hold the data and respond to external requests for data access.

**Funding:** The author(s) received no specific funding for this work.

**Competing interests:** The authors have declared that no competing interests exist.

those whose parents had no formal education were more likely to experience moderate to severe depression and anxiety than their counterparts. The ordinal logistic regression revealed that adolescents whose fathers had no formal education were 2.77 times more likely to experience depression (95% CI = 1.21–6.35, p < 0.05). Besides depression was more likely to be associated with those who were businessmen (OR = 1.95; 95% CI = 1.02–3.72, p < 0.05), and day laborers (OR = 0.41; 95% CI = 0.20–0.82, p < 0.05) and living with anxiety (OR = 0.05; 95% CI = 0.01–0.24, p <.000). Anxiety is prevalent among adolescents who had no formal education (OR = 0.15; 95% CI = 0.03–0.75, p < 0.05), no family savings (OR = 0.42; 95% CI = 0.23–0.78, p < 0.05) and depression (OR = 0.06; 95% CI = 0.01–0.22, p < 0.05).

## Conclusions

The findings of this study emphasize the importance of screening individuals for depressive and anxiety disorder symptoms to reduce the prevalence of mental health disorders among adolescents. In addition to screening, policymakers should incorporate policies to increase access to mental health services and promote mental health education for parents and the community.

## Introduction

Adolescence is a transitional period from childhood to adulthood and are at risk of various mental health disorders because of experiencing physical, psychosocial, and social changes during this period [1–4]. Mental health disorders have become a widespread public health concern, affecting around 25% of people globally [5]. According to the World Health Organization (WHO), depression and anxiety are among the leading causes of illness and disability among adolescents [6]. These conditions negatively impact their later lives [7] which can affect adolescents' development including their social functioning, family relationships, academic performance, school dropouts, and increased risk of tobacco use, pregnancy, mental health problems, and suicide [8–10].

Mental health disorder is still a severely under-recognized public health concern in Bangladesh. An estimation by the WHO reported that the prevalence of mental ailments in Bangladesh is 18.7% among adults and 12.6% among children [5,11]. National Mental Health Survey 2019 provided that the prevalence of depressive disorders is 6.7% and anxiety disorders is 4.7% in Bangladesh. WHO also estimates that the adolescents of Bangladesh, 10.2 percent of its total population are suffering from suicidal behavior, loneliness, anxiety, lack of close friends, social isolation, and substance use [2]. Depression is one of the most important causes of morbidity and disability in developing countries [3,12]. Bangladesh ranks fourth position in terms of the percentage of the population with depression and third position in terms of the number of cases of depression in the Southeast Asia region [12]. In Bangladesh, a few research have been conducted on the epidemiology of mental disorders among

adolescents [2,3,10,12,13]. A very recent study reported that the prevalence of depression and anxiety was 26.5 percent and 18.1 percent, respectively among school-going adolescents in Bangladesh [2]. Another study found 30.1 percent of school-going adolescents in urban, semi-urban, and rural areas in Bangladesh have depressive symptoms [10].

Strict parenting, parental separation, socio-economic conditions, violence (especially sexual violence and bullying), and drug addiction enhance the risk of depression among adolescents [3,9,10]. A systematic review article demonstrated that online social media also impacts adolescents' mental well-being in this digital era [9]. Previous studies have found several contributing socio-demographic and lifestyle factors to depression and anxiety such as sex, age, lower socio-economic condition, academic performance, parental educational attainment, living with the family, substance abuse, and satisfaction with sleep [2,14–17].

Although adolescents' mental health well-being is an integral part of attaining sustainable development goals, it remains a neglected area in the healthcare facilities and health policy of Bangladesh [3]. Hence, there is a need for early and effective identification of depression and anxiety that can prevent many mental disorders at an early age. However, available research mainly focuses on depression and anxiety in 10–19 years age group adolescents, school-going adolescents, or just female students [2,3,10,12,13]; ignoring adolescents who are out of school, live in rural areas, aged more than 19, and gender inclusiveness. Given the backdrop, this study aims to (1) estimate the prevalence of depression and anxiety among late adolescents aged 15–21 in Mymensingh, and (2) identify socio-demographic factors associated with these mental health conditions.

## Conceptual framework

This study was guided by a conceptual framework (**Fig 1**) demanding a hierarchical relationship between factors associated with depression and anxiety among adolescents in this setting. This model evolved from a review of the literature on depression and anxiety [2–4,10,18–21] and the experience of working with adolescents.

The first frame refers to the socio-demographic factors of study respondents such as age, gender, marital status, level of education, level of father's education, area of residence, occupation, household head, family monthly income, expenditure, and savings. We then added how socio-demographic factors are associated with depression and anxiety among adolescents.

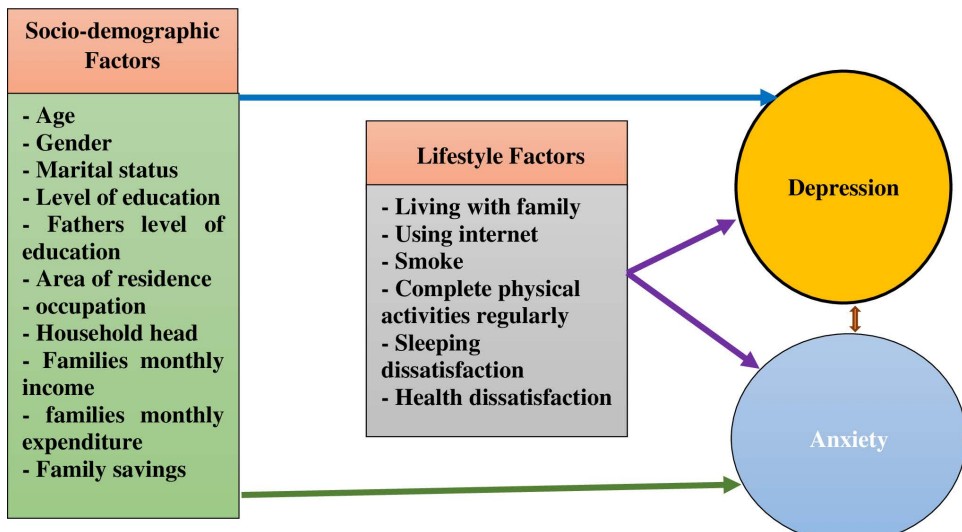

**Fig 1. Conceptual framework based on factors associated with depression and anxiety of adolescents.**

The second frame refers to the lifestyle-related factors of the respondents which include living with family, using the internet, smoking, completing regular physical activities, sleeping, and health dissatisfaction. Finally, we find out how depression accounts for adolescents' anxiety and how adolescents' anxiety also accounts for depression (**Fig 1**).

## Methodology

### Study design, setting, and participants

A cross-sectional study was conducted among late adolescents aged 15–21 years at Mymensingh division in Bangladesh following the quantitative data collection method. Initially, a multi-stage sampling technique was employed to reach the target population for the study. All the districts (Netrokona, Mymensingh, Sherpur, and Jamalpur) of Mymensingh were included as the data collection areas of this study. In the second stage, nineteenth Upazilas from these districts were selected randomly (**Fig 2**). In the Third stage, unions/pourashavas and villages/Mahallas were selected from each upazila. At the final stage respondents from each village were recruited conveniently using the inclusion criteria. Finally, 400 adolescents were successfully interviewed utilizing questionnaires through face-to-face interviews. The willingness of adolescents between 15 and 21 years old living in the Mymensingh division to participate in the survey was the inclusion criteria.

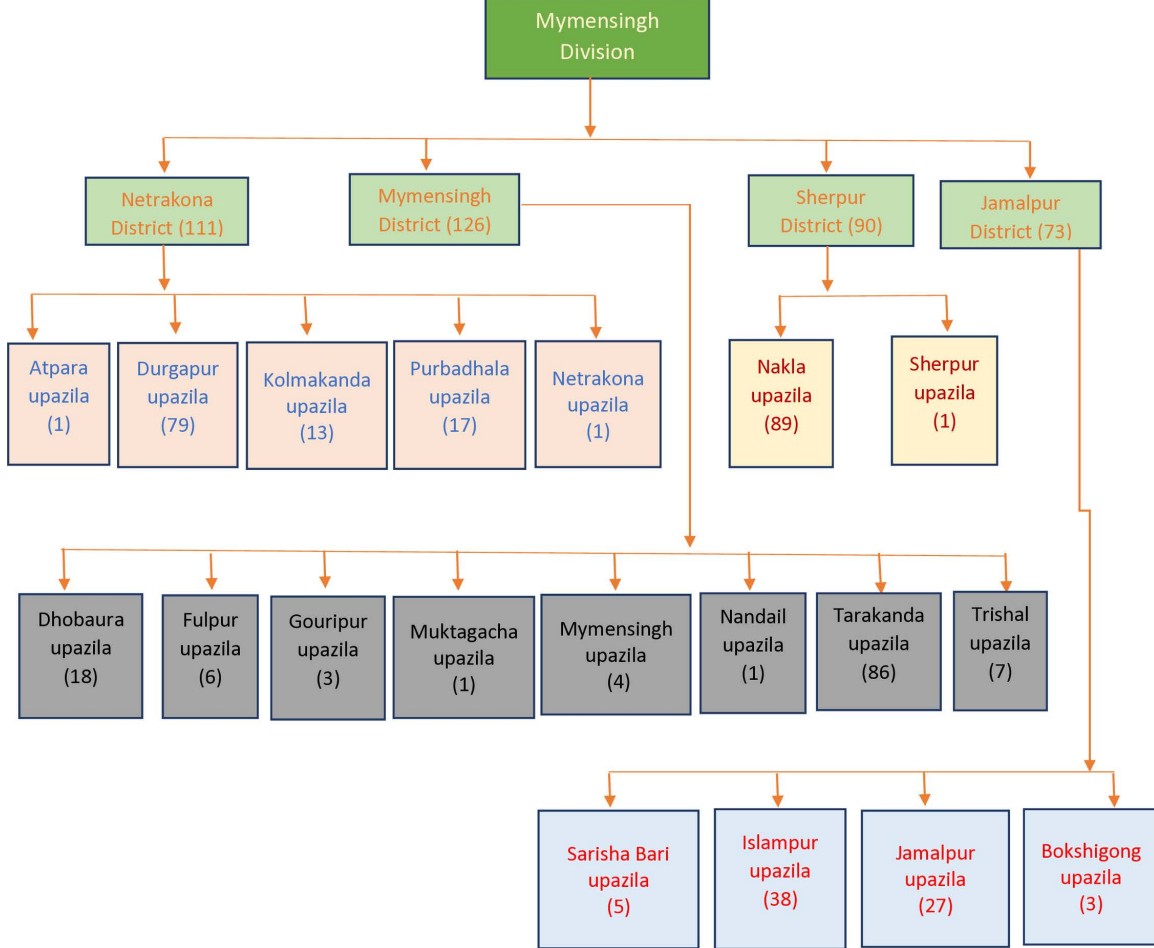

**Fig 2. Sampling technique used to select the study sample.**

Adolescents whose ages were under 15 and over 21 were excluded from data collection and the difficulty of completing the whole questionnaire was an exclusion criterion because their inclusion might create ambiguity to measure depression and anxiety among adolescents. Besides, adolescents currently receiving treatment for mental health conditions or those with cognitive impairments that hinder accurate self-reporting were excluded from this study. Four groups of highly educated investigators, each consisting of one male and three females were trained to collect data. This study collected data from August 1, 2022, to August 30, 2022. This study collected data from household visits so that all kinds of adolescents, i.e., out-of-school adolescents can be added.

The sample size was calculated using the following equation-

$$n = \frac{Z^2 \times pq}{e^2} \times 1$$

$$= \frac{(1.96)^2 \times 0.5(1-0.5)}{(0.05)^2} \times 1$$
$$= 384$$

Here, n = sample size
P = proportion of adolescents
q = 1−p
z = Standard normal deviation set at 1.96 corresponding to 95% confidence interval
e = Degree of accuracy desired = 5%

In the most recent study, [2] reported a prevalence rate of depression of 26.5% and anxiety of 18.1% focusing on school-going adolescents in Dhaka city in Bangladesh. In this study, the calculated sample was to be 384. However, 400 participants were recruited to ensure adequate power for the study. We could not find out the exact proportion of adolescents. So, we calculate sample size using (p=0.5).

## Measures

All the variables we used in this study were based on available literature on adolescent mental health in Bangladesh and similar contexts. Previous studies identified socioeconomic variables (age, sex, education, parental education, household income, and residence); lifestyle variables (mobile phone use, sleep satisfaction, physical activity, and smoking); and family structure as determinant factors for adolescent anxiety and depression. The variables we've included in this study are discussed briefly below.

**Socio-demographic measures.** Socio-demographic data were collected, including age, sex, religion, marital status, respondents' level of education, respondents' fathers and mothers' level of education, household head (yes/no), occupation, monthly family income, monthly family expenditure, family savings (yes/no) and living area (urban/rural).

**Lifestyle-related variables.** The lifestyle-related data were collected by asking questions regarding live with family (yes/no), regular physical activities (yes/no), use of the internet (yes/no), sleep satisfaction (yes/no), health satisfaction (yes/no), and smoking (yes/no) [8].

**Patient Health Questionnaire (PHQ-9).** The nine-item Bangla version Patient Health Questionnaire (PHQ-9) scale was used to assess the depressive symptoms among the participants [22]. This scale consists of 9 questions with a four-point Likert scale ranging from 0 ("Never") to 3 ("Always") [23]. The level of depression was categorized into five groups minimal, mild, moderate, moderately severe, and severe based on scoring in the range of 0–4, 5–9, 10–14, 15–19, and >20, respectively (1). In the present study, those scoring in the moderate to severe range were classed as depression positive. In the present study, the PHQ-9 scale was found to have good reliability (Cronbach's α = 0.87) greater than the recommended value of 0.70, indicating excellent reliability.

**Generalized Anxiety Disorder (GAD-7).** The Generalized Anxiety Disorder (GAD-7) scale, which has seven items in Bangla, was utilized to evaluate the individuals' anxiety issues [24]. The scale consists of 7-item questions with a four-point Likert scale ranging from 0 ("Not at all") to 3 ("Always") [25]. The level of anxiety was categorized into four groups minimal, mild, moderate, and severe based on scoring in the range of 0–4, 5–9, 10–14, and >15, respectively (1). In this study, those scoring in the moderate to severe range were classified as having anxiety disorder. The value of the reliability coefficient Cronbach's α for the overall GAD-7 scale was 0.85 which is greater than the recommended value 0.70, indicating excellent reliability.

The PHQ-9 and GAD-7 were translated into Bangla for use in the Bangladeshi context. Previous studies in Bangladesh have shown these tools to be valid, with high internal consistency (Cronbach's alpha: PHQ-9 = 0.79–0.84; GAD-7 = 0.80–0.85) and robust psychometric properties across diverse populations, including adolescents [2,3,10]. These validations confirm their appropriateness for diagnosing mental health conditions within the local cultural and socio-demographic context.

Adolescent mental health is influenced by several factors, including a family history of depression or anxiety, educational outcomes, financial stress, household wealth, physical health indicators (e.g., BMI), drug addiction, frustration about life or love, etc. However, these factors were excluded for the following reasons:

**Family history of depression or anxiety.** Due to stigma and lack of official diagnoses, family histories of anxiety and depression are frequently underreported in Bangladesh [2,4,6]. Furthermore, it was thought to be difficult for teenagers to accurately self-report on the mental health of their parents.

**Educational outcomes.** Although academic stress may play a role, our research concentrated on more general sociodemographic factors. Parental educational attainment was used to measure the indirect impact of education.

**Financial stress, and wealth.** We employed monthly family income, spending, and savings as a proxy for economic status in place of explicit financial stress measures, as these have been often employed in related research [12].

**Physical health indicators (e.g., BMI).** Physical health indicators, e.g., BMI were excluded from data collection due to practical limitations in measuring anthropometric data in field-based surveys. However, we employed self-reported health satisfaction as a proxy following similar studies [10].

**Drug addiction.** We avoided difficult items that could result in underreporting because of ethical considerations and the self-reported nature of our study. Nonetheless, one relevant lifestyle component was smoking status.

**Frustration about Life or Love.** Although psychological stressors are significant, our study concentrated on behavioral and sociodemographic factors that have been validated in prior research and can be quantified objectively.

## Statistical analysis

Data entry, editing, and analysis were performed using the Statistical Package for the Social Sciences (SPSS) software program, version 26.0. We used descriptive statistics (e.g., frequencies, percentages, mean, standard deviation) and first-order analysis (i.e., chi-square tests). Variables that had a p-value of less than or equal to 0.05 were considered statistically significant and further analyzed using ordinal logistic regression. Although multinomial logit and ordinal logistic regression are used for categorical data with more than two categories. The dependent variable of our study, depression and anxiety level, were measured in ordinal scale (Minimal, mild, moderate, moderately severe, and severe) so we employed ordinal logistic regression (OLR). OLR is suitable when the outcome variable is categorically ordered and allows us to estimate the probability of falling into a higher category of anxiety or depression while several predictor variables are accounted. Due to the categorical nature of depression and anxiety outcome, an ordinal logistic regression model with robust variance was used to investigate the relationship between socio-demographic and lifestyle factors and depression and anxiety of adolescents. The robust variance estimator at the correlation level approximates a comparable Generalized Estimating Equations (GEE) ordinal model.

**Ordinal Logistic Regression Model.** Based on the cumulative odds technique, the ordinal logistic regression model makes the proportional odds assumption about the link between independent factors and the ordinal response variable. The following is a representation of the model:

$$Log(\frac{P(Y \leq j)}{P(Y > j)} = \beta0 + \beta1X1 + \beta2X2 + \ldots + BkXk$$

**Here,**

P (Y≤j) is the cumulative probability of the outcome variable being at or below category j,

P(Y>j) is the probability of being above category j,

$\beta_0$ is the intercept,

$X_1$, $X_2$,..., $X_k$ are the independent variables (e.g., age, gender, parental education, family income, physical activity, smoking),

$\beta_1$, $\beta_2$,..., $\beta_k$ are the regression coefficients representing the change in the log-odds of being in a higher category for a one-unit increase in the predictor variable.

The Omnibus Test of Model Coefficients gives an overall indication of the "goodness of fit test" and reports that our ordinal logistic regression model performed well and would be a good predictor of the two outcome variables (depression and anxiety). Model fits for predicting depression among adolescents and their different socio-demographic and lifestyle variables $x^2$ = 54.42 ($p < 0.001$) and Nagelkerke $R^2$ = 0.14; anxiety among adolescents and their different socio-demographic and lifestyle variables $x^2$ = 67.37 ($p < 0.001$) and Nagelkerke $R^2$ = 0.18. According to the proportional odds assumption, there is a constant correlation between predictors and outcome odds at every cumulative level of the dependent variable. Following best practices in public health research, we first reduced overfitting and multicollinearity by including only variables with $p < 0.05$ from the bivariate analysis in the ordinal logistic regression [6,20,26].

## Ethical considerations

All procedures of the study are approved by the Higher Degree Academic Research Approval Committee, Department of Population Science, Jatiya Kabi Kazi Nazrul Islam University, Trishal, Mymensingh-2224, Bangladesh, approval number (JKKNIU/DPS/AC/EA/2024/245). A written informed consent form was provided to each respondent of the study including the study purpose, potential risks, and a confidentiality and anonymity statement of how their given information would be securely handled. Moreover, the informed consent form also acknowledged that the respondent could feel free to leave the interview at any time of the interview if he/she feels discomfort. Signed was collected in the written consent from all the study respondents before starting the interviews. Besides, written consent was obtained from the guardians of the respondents aged below 18. Any deception or exaggeration about the aims and objectives of this research was avoided, and any type of communication about the research was done with honesty and transparency.

## Results

This cross-sectional study consisted of 400 participants where 62.3% were male and 37.8% were female, and their mean age was 18.26 years (SD ±1.82) (Table 1). 55.3% of the study respondents were from families with income of 15000–30000 per month and 85% lived with their families in rural areas (89.3%). Most of the respondents were Muslim (93.5%), unmarried (79.8%), students (51.5%), had secondary education (53.8%), without formal education of fathers (42.5%) and mothers (41.8%), not household head (78.8%), and their inability to save money each month (17.3%), uses mobile phone (79.0%), use the internet (65.0%), maintained regular physical activities (78.3%), satisfied with their sleep (73.3%), satisfied with their health (61.8%) and non-smokers (85.3%).

**Table 1. Socio-demographic profile of the respondents.**

| variables | | Frequency | Percentage (%) |
|---|---|---|---|
| Age | 15-16 | 82 | 20.5 |
| | 17-18 | 129 | 32.3 |
| | 19-21 | 189 | 47.3 |
| Gender | Male | 249 | 62.3 |
| | Female | 151 | 37.8 |
| Religion | Muslim | 374 | 93.5 |
| | Non-Muslim | 26 | 6.5 |
| Marital Status | Married | 81 | 20.3 |
| | Unmarried | 319 | 79.8 |
| Level of respondent's education | Without formal education | 10 | 2.5 |
| | Primary | 55 | 13.8 |
| | Secondary | 215 | 53.8 |
| | Higher Secondary | 82 | 20.5 |
| | Undergraduate | 38 | 9.5 |
| Fathers' level of education | Without formal education | 170 | 42.5 |
| | Primary | 117 | 29.3 |
| | Secondary | 86 | 21.5 |
| | Higher secondary or above | 27 | 6.8 |
| Mother's level of education | Without formal education | 167 | 41.8 |
| | Primary | 136 | 34.0 |
| | Secondary | 88 | 22.0 |
| | Higher secondary or above | 9 | 2.3 |
| Total family members of the respondents | 2-5 | 216 | 54.0 |
| | >5 | 184 | 46.0 |
| Respondent's occupation | Unemployed | 30 | 7.5 |
| | Business/Small business | 42 | 10.5 |
| | Job | 40 | 10.0 |
| | Housewife | 41 | 10.3 |
| | Day laborer | 41 | 10.3 |
| | Students | 206 | 51.5 |
| Respondent's monthly family income | <15000 | 143 | 35.8 |
| | 15000-30000 | 221 | 55.3 |
| | >30000 | 36 | 9.0 |
| Respondent's monthly family expenditure | <10000 | 86 | 21.5 |
| | 10000-20000 | 248 | 62.0 |
| | >20000 | 66 | 16.5 |
| Respondent's family savings | Yes | 331 | 82.8 |
| | No | 69 | 17.3 |
| Area of residence | Rural | 357 | 89.3 |
| | Urban | 43 | 10.8 |
| Using mobile phone | Yes | 316 | 79.0 |
| | No | 84 | 21.0 |
| Living with family | Yes | 340 | 85.0 |
| | No | 60 | 15.0 |
| Regular physical activity | Yes | 313 | 78.3 |
| | No | 87 | 21.8 |

*(Continued)*

**Table 1.** (Continued)

| variables | | Frequency | Percentage (%) |
|---|---|---|---|
| Smoker | Yes | 59 | 14.8 |
| | No | 341 | 85.3 |
| Internet use | Yes | 260 | 65.0 |
| | No | 140 | 35.0 |
| Sleeping satisfaction | Yes | 293 | 73.3 |
| | No | 107 | 26.8 |
| Health satisfaction | Yes | 248 | 62.0 |
| | No | 152 | 38.0 |
| Household head | Yes | 85 | 21.3 |
| | No | 315 | 78.8 |

We found an overall prevalence of depression to be 47.8%. Based on the PHQ-9 scale, results indicated that minimal, mild, moderate, moderately severe, and severe depression levels were present in 30 (7.5%), 179 (44.8%), 134 (33.5%), 42 (10.5%), and 15 (3.8%), respectively (Fig 3).

Table 2 shows that adolescents whose fathers have no formal education suffer from depression compared to their counterparts and this finding is statistically significant ($x^2$ = 21.026; p = 0.05). This study found a significant ($x^2$ = 50.041; p< 0.001) relationship between respondents' occupation and depression. The students were more depressed than respondents who were in business, jobs, housewives, and day laborers. As the number of types of exposure to mobile phones, the percentage of adolescents with depression increased from non-mobile phone users. The adolescents who lived with family were suffering from more depressive symptoms than adolescents who didn't live with their family and this result is statistically associated ($x^2$ = 10.541; p = 0.032). In terms of lifestyle variables, among the total adolescents who were found to have depressive symptoms, those adolescents who maintained regular physical activities and satisfactory sleep had more depression than their counterparts. Again, adolescent's health satisfaction was found to be statistically significant ($x^2$ = 14.083; p = 0.007) with depressive symptoms. No significant interactions were found with age, gender, marital status, educational attainment, monthly family income, monthly family expenditure, and internet use and we had

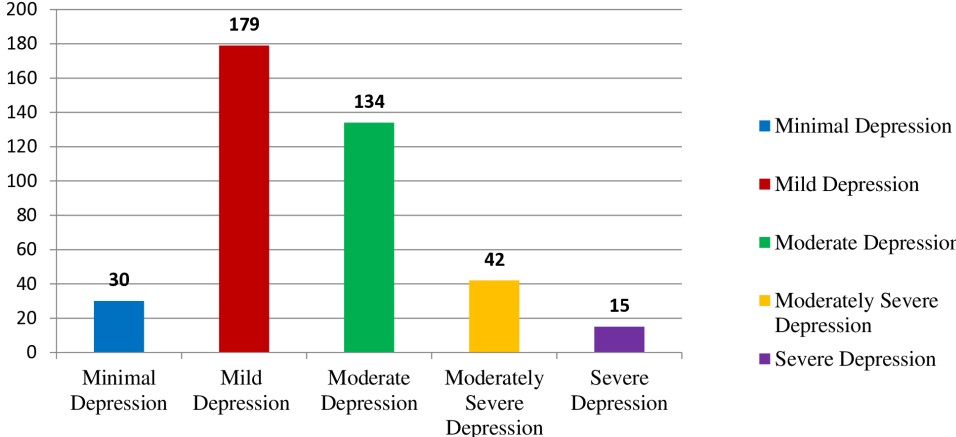

**Fig 3. Prevalence and severity of depression.**

 

**Table 2. Distribution of variables and association with depression among adolescents.**

| Variables | Depression | | | | | Total N (%) | $\chi^2$ (P value) |
|---|---|---|---|---|---|---|---|
| | Minimal depression N (%) | Mild depression N (%) | Moderate depression N (%) | Moderately severe depression N (%) | Severe depression N (%) | | |
| | Total Depression: 47.8% | | | | | | |
| **Fathers' level of education** | | | | | | | |
| Without formal education | 7 (1.8) | 73 (18.3) | 64 (16.0) | 19 (4.8) | 7 (1.8) | 170 (42.5) | 21.026 (.050) |
| Primary | 10 (2.5) | 56 (14.0) | 41 (10.3) | 9 (2.3) | 1 (0.3) | 117 (29.3) | |
| Secondary | 9 (2.3) | 34 (8.5) | 27 (6.8) | 10 (2.5) | 6 (1.5) | 86 (21.5) | |
| Higher secondary or above | 4 (1.0) | 16 (4.0) | 2 (0.5) | 4 (1.0) | 1 (0.3) | 27 (6.8) | |
| **Respondent's occupation** | | | | | | | |
| Unemployed | 2 (0.5) | 11 (2.8) | 12 (3.0) | 2 (0.5) | 3 (0.8) | 30 (7.5) | 50.041 (.000) *** |
| Business/Small business | 2 (0.5) | 11 (2.8) | 18 (4.5) | 9 (2.3) | 2 (0.5) | 42 (10.5) | |
| Job | 2 (0.5) | 19 (4.8) | 9 (2.3) | 8 (2.0) | 2 (0.5) | 40 (10.0) | |
| Housewife | 1 (0.3) | 22 (5.5) | 17 (4.3) | 1 (0.3) | 0 (0.0) | 41 (10.3) | |
| Day laborer | 8 (2.0) | 19 (4.8) | 7 (1.8) | 2 (0.5) | 5 (1.3) | 41 (10.3) | |
| Students | 15 (3.8) | 97 (24.3) | 71 (17.8) | 20 (5.0) | 3 (0.8) | 206 (51.5) | |
| **Respondent's family savings** | | | | | | | |
| Yes | 27 (6.8) | 147 (36.8) | 114 (28.5) | 35 (8.8) | 8 (2.0) | 331 (82.8) | 10.765 (.029)* |
| No | 3 (0.8) | 32 (8.0) | 20 (5.0) | 7 (1.8) | 7 (1.8) | 69 (17.3) | |
| **Using mobile phone** | | | | | | | |
| Yes | 20 (5.0) | 138 (34.5) | 115 (28.7) | 34 (8.5) | 9 (2.3) | 316 (79.0) | 10.261 (.036)* |
| No | 10 (2.5) | 41 (10.3) | 19 (4.8) | 8 (2.0) | 6 (1.5) | 84 (21.0) | |
| **Living with family** | | | | | | | |
| Yes | 25 (6.3) | 146 (36.5) | 123 (30.8) | 36 (9.0) | 10 (2.5) | 340 (85.0) | 10.541 (.032)* |
| No | 5 (1.3) | 33 (8.3) | 11 (2.8) | 6 (1.5) | 5 (1.3) | 60 (15.0) | |
| **Regular physical activity** | | | | | | | |
| Yes | 22 (5.5) | 151 (37.8) | 104 (26.0) | 26 (6.5) | 10 (2.5) | 313 (78.3) | 12.157 (.016)* |
| No | 8 (2.0) | 28 (7.0) | 30 (7.5) | 16 (4.0) | 5 (1.3) | 87 (21.8) | |
| **Sleeping satisfaction** | | | | | | | |
| Yes | 29 (7.2) | 131 (32.8) | 95 (23.8) | 27 (6.8) | 11 (2.8) | 293 (73.3) | 10.497 (.033)* |
| No | 1 (0.3) | 48 (12.0) | 39 (9.8) | 15 (3.8) | 4 (1.0) | 107 (26.8) | |
| **Health satisfaction** | | | | | | | |
| Yes | 22 (5.5) | 122 (30.5) | 77 (19.3) | 17 (4.3) | 10 (2.5) | 248 (62.0) | 14.083 (.007) ** |
| No | 8 (2.0) | 57 (14.2) | 57 (14.2) | 25 (6.3) | 5 (1.3) | 152 (38.0) | |
| **Anxiety** | | | | | | | |
| Minimal | 9 (2.3) | 15 (3.8) | 6 (1.5) | 2 (0.5) | 2 (0.5) | 34 (8.5) | 49.225 (.000)*** |
| Mild | 21 (5.3) | 110 (27.5) | 78 (19.5) | 21 (5.3) | 5 (1.3) | 235 (58.8) | |
| Moderate | 0 (0.0) | 53 (13.3) | 47 (11.8) | 18 (4.5) | 6 (1.5) | 124 (31.0) | |
| Severe | 0 (0.0) | 1 (0.3) | 3 (0.8) | 1 (0.3) | 2 (0.5) | 7 (1.8) | |

*** =significant at 0.001 level, ** = significant at 0.01 level and * = significant at 0.05 level

discussed them in the discussion part too. There was evidence of a significant association between depression and adolescents who were anxious.

The overall prevalence of anxiety was 32.8%. Based on the GAD-7 scale, results indicated that minimal, mild, moderate, and severe anxiety levels had rates of 8.5%, 58.8%, 31%, and 1.8%, respectively. Table 3 shows that most of the

**Table 3. Distribution of variables and association with anxiety among adolescents.**

| Variables | Anxiety | | | | Total N (%) | $\chi^2$ (P value) |
|---|---|---|---|---|---|---|
| | Minimal anxiety N (%) | Mild anxiety N (%) | Moderate anxiety N (%) | Severe anxiety N (%) | | |
| | Total Anxiety: 32.8% | | | | | |
| **Level of respondent's education** | | | | | | |
| Without formal education | 6 (1.5) | 2 (0.5) | 2 (0.5) | 0 (0.0) | 10 (2.5) | 48.624 (.000)*** |
| Primary | 5 (1.3) | 39 (9.8) | 11 (2.8) | 0 (0.0) | 55 (13.8) | |
| Secondary | 13 (3.3) | 118 (29.5) | 80 (20.0) | 4 (1.0) | 215 (53.8) | |
| Higher Secondary | 6 (1.5) | 54 (13.5) | 19 (4.8) | 3 (0.8) | 82 (20.5) | |
| Undergraduate | 4 (1.0) | 22 (5.5) | 12 (3.0) | 0 (0.0) | 38 (9.5) | |
| **Fathers' level of education** | | | | | | |
| Without formal education | 22 (5.5) | 101 (25.3) | 47 (11.8) | 0 (0.0) | 170 (42.5) | 37.620 (.000)*** |
| Primary | 4 (1.0) | 72 (18.0) | 41 (10.3) | 0 (0.0) | 117 (29.3) | |
| Secondary | 4 (1.0) | 48 (12.0) | 27 (6.8) | 7 (1.8) | 86 (21.5) | |
| Higher secondary or above | 4 (1.0) | 14 (3.5) | 9 (2.3) | 0 (0.0) | 27 (6.8) | |
| **Respondent's monthly family expenditure** | | | | | | |
| <10000 | 15 (3.8) | 42 (10.5) | 29 (7.2) | 0 (0.0) | 86 (21.5) | 21.750 (.001)*** |
| 10000-20000 | 13 (3.3) | 161 (40.3) | 70 (17.5) | 4 (1.0) | 248 (62.0) | |
| >20000 | 6 (1.5) | 32 (8.0) | 25 (6.3) | 3 (0.8) | 66 (16.5) | |
| **Respondent's family savings** | | | | | | |
| Yes | 22 (5.5) | 209 (52.3) | 95 (23.8) | 5 (1.3) | 331 (82.8) | 17.956 (.000)*** |
| No | 12 (3.0) | 26 (6.5) | 29 (7.2) | 2 (0.5) | 69 (17.3) | |
| **Area of residence** | | | | | | |
| Rural | 30 (7.5) | 214 (53.5) | 110 (27.5) | 3 (0.8) | 357 (89.3) | 16.583 (.001)*** |
| Urban | 4 (1.0) | 21 (5.3) | 14 (3.5) | 4 (1.0) | 43 (10.8) | |
| **Using mobile phone** | | | | | | |
| Yes | 22 (5.5) | 184 (46.0) | 105 (26.3) | 5 (1.3) | 316 (79.0) | 6.908 (.075) |
| No | 12 (3.0) | 51 (12.8) | 19 (4.8) | 2 (0.5) | 84 (21.0) | |
| **Regular physical activity** | | | | | | |
| Yes | 18 (4.5) | 195 (48.8) | 96 (24.0) | 4 (1.0) | 313 (78.3) | 17.766 (.000)*** |
| No | 16 (4.0) | 40 (10.0) | 28 (7.0) | 3 (0.8) | 87 (21.8) | |
| **Smoker** | | | | | | |
| Yes | 11 (2.8) | 26 (6.5) | 22 (5.5) | 0 (0.0) | 59 (14.8) | 13.012 (.005)** |
| No | 23 (5.8) | 209 (52.3) | 102 (25.5) | 7 (1.8) | 341 (85.3) | |
| **Sleeping satisfaction** | | | | | | |
| Yes | 30 (7.5) | 183 (45.8) | 75 (18.8) | 5 (1.3) | 293 (73.3) | 16.784 (.001)*** |
| No | 4 (1.0) | 52 (13.0) | 49 (12.3) | 2 (0.5) | 107 (26.8) | |
| **Health satisfaction** | | | | | | |
| Yes | 31 (7.8) | 157 (39.3) | 59 (14.8) | 1 (0.3) | 248 (62.0) | 32.298 (.000)*** |
| No | 3 (0.8) | 78 (19.5) | 65 (16.3) | 6 (1.5) | 152 (38.0) | |
| **Depression** | | | | | | |
| Minimal | 9 (2.3) | 21 (5.3) | 0 (0.0) | 0 (0.0) | 30 (7.5) | 49.225 (.000)*** |
| Mild | 15 (3.8) | 110 (27.5) | 53 (13.3) | 1 (0.3) | 179 (44.8) | |
| Moderate | 6 (1.5) | 78 (19.5) | 47 (11.8) | 3 (0.8) | 134 (33.5) | |
| Moderately severe | 2 (0.5) | 21 (5.3) | 18 (4.5) | 1 (0.3) | 42 (10.5) | |
| Severe | 2 (0.5) | 5 (1.3) | 6 (1.5) | 2 (0.5) | 15 (3.8) | |

*** = significant at 0.001 level, ** = significant at 0.01 level and * = significant at 0.05 level

adolescents with a secondary level education were more anxious than other levels of education. Respondents whose father's level of educational qualification was reported to be without formal education are more anxious than others and this finding is statistically significant ($x^2$ = 37.620; p< 0.001). Besides, those adolescent families with middle-level monthly expenditures (10000–20000) had more anxiety than their counterparts. In terms of residential areas, adolescents living in rural areas were significantly more likely to be anxious than those living in urban areas and this result is statistically significant ($x^2$ = 16.583; p< 0.001). Besides, adolescent's health satisfaction was found to be statistically significant ($x^2$ = 32.298; p< 0.001) with anxiety. No significant interactions were found with age, gender, marital status, occupation, monthly family income, and household head. There was a statistically significant association between anxiety and adolescents who were depressed.

The results of ordinal logistic regression analyses are presented in Tables 4 and 5. The results indicate that adolescents whose fathers had no formal education were significantly more likely to experience depression (OR = 2.77, 95% CI = 1.21–6.35, p = 0.016). Likewise, the risks of depression were lower for day laborer (OR = 0.41, 95% CI = 0.20–0.82, p = 0.012) and greater for businessmen (OR = 1.95, 95% CI = 1.02–3.72, p = 0.043). However, adolescents who completed

**Table 4. Ordinal logistic regression analyses of factors associated with depression among adolescents.**

**Depression of Adolescents**

| Variables | Categories | Parameter Estimates | Standard Error (S.E) | P-value | Odd Ratio | 95% CI | |
|---|---|---|---|---|---|---|---|
| | | | | | | Lower | Upper |
| Fathers' level of education | Without formal education | 1.019 | .423 | .016* | 2.771 | 1.210 | 6.346 |
| | Primary | .462 | .426 | .278 | 1.587 | .689 | 3.655 |
| | Secondary | .618 | .437 | .157 | 1.856 | .789 | 4.366 |
| | Higher secondary or above | – | – | – | 1.00 | – | – |
| Respondent's occupation | Unemployed | .407 | .383 | .288 | 1.502 | .709 | 3.183 |
| | Business/small business | .667 | .330 | .043* | 1.948 | 1.021 | 3.720 |
| | Job | .239 | .339 | .480 | 1.270 | .654 | 2.466 |
| | Housewife | -.260 | .347 | .453 | .771 | .391 | 1.521 |
| | Day laborer | -.894 | .357 | .012** | .409 | .203 | .822 |
| | Students | – | – | – | 1.00 | – | – |
| Family savings | Yes | -.436 | .270 | .106 | .647 | .381 | 1.098 |
| | No | – | – | – | 1.00 | – | – |
| Live with family | Yes | .357 | .294 | .225 | 1.429 | .803 | 2.541 |
| | No | – | – | – | 1.00 | – | – |
| Using mobile | Yes | .288 | .254 | .256 | 1.334 | .812 | 2.192 |
| | No | – | – | – | 1.00 | – | – |
| Complete regular physical activity | Yes | -.710 | .252 | .005** | .492 | .300 | .805 |
| | No | – | – | – | 1.00 | – | – |
| Sleeping satisfaction | Yes | -.153 | .233 | .511 | .858 | .543 | 1.355 |
| | No | – | – | – | 1.00 | – | – |
| Health satisfaction | Yes | -.321 | .219 | .144 | .726 | .472 | 1.115 |
| | No | – | – | – | 1.00 | – | – |
| Anxiety | Minimal | -3.078 | .834 | .000*** | .046 | .009 | .236 |
| | Mild | -2.028 | .752 | .007** | .132 | .030 | .575 |
| | Moderate | -1.505 | .751 | .045 | .222 | .051 | .967 |
| | Severe | – | – | – | 1.00 | – | – |

*** = significant at 0.001 level, ** = significant at 0.01 level, * = significant at 0.05 level and CI = Confidence Interval

**Table 5. Ordinal logistic regression analyses of factors associated with anxiety among adolescents.**

Anxiety of Adolescents

| Variables | Categories | Parameter Estimates | Standard Error (S.E) | P-value | Odd Ratio | 95% CI | |
|---|---|---|---|---|---|---|---|
| | | | | | | Lower | Upper |
| Respondent's level of education | Without formal education | -1.916 | .831 | .021* | .147 | .029 | .751 |
| | Primary | .153 | .470 | .745 | 1.165 | .464 | 2.927 |
| | Secondary | .503 | .383 | .189 | 1.654 | .781 | 3.505 |
| | Higher Secondary | .144 | .425 | .736 | 1.154 | .502 | 2.656 |
| | Undergraduate | – | – | – | 1.00 | – | – |
| Fathers' level of education | Without formal education | -.218 | .469 | .643 | .804 | .321 | 2.019 |
| | Primary | .399 | .475 | .400 | 1.491 | .588 | 3.782 |
| | Secondary | .656 | .474 | .166 | 1.927 | .762 | 4.875 |
| | Higher secondary or above | – | – | – | 1.00 | – | – |
| Family savings | Yes | -.857 | .310 | .006** | .424 | .231 | .779 |
| | No | – | – | – | 1.00 | – | – |
| Respondent's monthly family expenditure | <10000 | -.706 | .386 | .067 | .493 | .232 | 1.051 |
| | 10000-20000 | -.276 | .316 | .382 | .759 | .409 | 1.409 |
| | >20000 | – | – | – | 1.00 | – | – |
| Area of residence | Rural | -.473 | .363 | .192 | .623 | .306 | 1.269 |
| | Urban | – | – | – | 1.00 | – | – |
| Smoke | Yes | .121 | .326 | .712 | 1.128 | .595 | 2.139 |
| | No | – | – | – | 1.00 | – | – |
| Sleeping satisfaction | Yes | -.595 | .255 | .020* | .551 | .335 | .909 |
| | No | – | – | – | 1.00 | – | – |
| Health satisfaction | Yes | -.907 | .234 | .000*** | .404 | .255 | .638 |
| | No | – | – | – | 1.00 | – | – |
| Depression | Minimal | -2.901 | .697 | .000*** | .055 | .014 | .215 |
| | Mild | -1.159 | .563 | .039* | .314 | .104 | .945 |
| | Moderate | -.732 | .568 | .197 | .481 | .158 | 1.463 |
| | Moderately severe | -.706 | .628 | .261 | .493 | .144 | 1.690 |
| | Severe | – | – | – | 1.00 | – | – |

*** = significant at 0.001 level, ** = significant at 0.01 level, * = significant at 0.05 level and CI =Confidence Interval

regular physical activities and who were satisfied with sleep and health were associated with lower odds of depression. We also see that respondents who were anxious severely were more likely to have depressive symptoms. Depression and anxiety were strongly associated with each other (Tables 4 and 5).

Ordinal logistic regression analyses of adolescent anxiety with all significant factors of the chi-square test are given in Table 5. The participants who smoke were 1.13 times more likely to be anguished than those who do not smoke (OR = 1.13; 95% CI = 0.60–2.14). Likewise, associated factors of depression including whose father had no formal education (OR = 2.77; 95% CI = 1.21–6.35, p = 0.016), who had a business (OR = 1.95; 95% CI = 1.02–3.72, p = 0.043), day laborer (OR = 0.41; 95% CI = 0.20–0.82, p = 0.012), complete regular physical activities (OR = 0.49; 95% CI = 0.30–0.81, p = 0.005) and anxiety (OR = 0.05; 95% CI = 0.01–0.24, p <.000), whilst for anxiety, only association with being without formal education (OR = 0.15; 95% CI = 0.03–0.75, p = 0.021), family savings (OR = 0.42; 95% CI = 0.23–0.78, p = 0.006), sleeping satisfaction (OR = 0.55; 95% CI = 0.34–0.91, p = 0.20), health satisfaction (OR = 0.06; 95% CI = 0.26–0.64, p <.000) and depression (OR = 0.06; 95% CI = 0.01–0.22, p <.000) were retained in ordinal logistic analyses (Table 5).

## Discussion

This study examined the prevalence of, and factors associated with depression and anxiety among adolescents aged 15–21 years in Mymensingh division, Bangladesh. The findings of the present study indicated that nearly half of the respondents (47.8%) experienced moderate to severe depression. The prevalence of depression found in the present study was higher (47.8% vs 26.5%) than in previous study conducted in Dhaka city [2,10]. Conversely, the prevalence of depression in this study was found lower (47.8% vs 49%) compared to a study conducted in 2012 [13]. A variety of factors may influence the rates of depression and anxiety in the study findings. Rates may fluctuate across variances in socio-demographic characteristics, especially age groups, gender differences, and study area due to significant contextual effects in depression and anxiety. Besides, sample size is another reason for the variations in the study findings [2,10,13].

Additionally, this study revealed that 32.8% of respondents experienced moderate to severe anxiety whereas a prior study found 18.1% of adolescents had anxiety [2]. Moreover, several studies conducted in other Asian countries have reported high prevalence rates of depression and anxiety among adolescents, for example, Jeelani et al. (2022) stated that anxiety was 20% among adolescents in India, another study Jha et al. (2017) stated that depression is 25.8% among adolescents in India. The prevalence of anxiety and depression among school-going adolescents in Pakistan was 53.25% [27]. In Sri Lanka's prevalence of depression among adolescents was 39% [28] and anxiety in Malaysia was 29% [29]. This difference may occur due to the age of the respondents, gender and cultural differences among the studies.

It has been shown that the prevalence of depression and anxiety rises with age. Depression is most common in adolescents aged between 19–21 reported by this cross-sectional study where Islam et al. (2021) found that 15–16 years of age of adolescents had severe depression. Various factors are responsible for depressive symptoms and anxiety among adolescents aged 19–21 years including increased academic pressure, more time spent on screen-based sedentary behavior, and family pressure. During this time, adolescents lack proper support and warmth from parents, leading to depressive symptoms [10]. The study results revealed gender differences in depression, where male adolescents were more vulnerable than females. Anjum et al. (2021) and Derluyn et al. (2004) reported the opposite finding that the prevalence of depression was higher among female school students (60.8%) than male school students (39.2%). The findings are not aligned with other studies as other studies were conducted in institutional settings where all the respondents were students. On the contrary, this study included all adolescents despite their institutions and occupations. Several factors are responsible for the high prevalence of depressive symptoms and anxiety among male adolescents. The social structure puts extensive pressure directly or indirectly on male adolescents as they pass through different physiological and mental changes. Further, male adolescents had extra pressure for household responsibilities at an early age. Thus, male adolescents are anguished to smoke, and excessive use of the internet. These are some of the pressing and practical factors through which male adolescents of low and middle-income countries like Bangladesh go through, and these factors might play an important role behind the high prevalence of depressive symptoms among them [10].

Moreover, the prevalence of depression and anxiety scores was higher among unmarried adolescents because they are much more involved with social media, worried about choosing a life partner, and worried about different health issues and study, previous study also reported similar findings [18]. Our results revealed an association of anxiety scores with respondents who live in rural areas, which is contrary with previous findings [10]. This variation has been explained to be due to the study design and study settings. The reason behind this could be negative life experiences, low parental warmth (high levels of maternal aggression and escalating adolescent-parent conflict), physical inactivity and lifestyle habits [10].

In addition, the current study results also revealed a strong association between adolescents' depression with their fathers' level of education. It was reported that depression is more common among those whose fathers had no formal education. A previous study showed a wide variation indicating that respondents whose fathers had higher levels of education have more depressive symptoms [10]. The reason for this variation may be due to the selection of a larger sample of respondents from rural areas than from urban areas whereas Anjum et al. (2021) selected a sample from urban adolescents.

The association between occupation of adolescents and depression is uncommon. These findings demonstrated that 51.5% of the participants were students and they were found to have depressive symptoms. This observation is supported by previous studies[2,3,6,9,10,20,30–33]; reported higher levels of depression among adolescent students but as the regression analysis showed businessmen were 1.95 times and adolescents who work as day laborers were 0.41 times more likely to be suffering from depressive symptoms than students and it is statistically significant (Table 4). At an early age, they must go to work without studying, and where they face a lot of harassment too. This result was not consistent with the findings of other studies because they conducted their study only in institutional settings.

Living with family, using the internet, getting little sleep, and having poor health were the main risk factors for depression. The results showed that adolescents who did not get enough sleep were depressed, which was consistent with earlier research from Bangladesh [2]. Adolescents living with family frequently notice that their family is financially unstable, disputes in the family, and some families give greater significance to the siblings, which is consistent with previous research [2,20]. Internet use also affects the sleep habits of adolescents found in this study and is also aligned with findings from prior research [10,34].

The prevalence of anxiety scores was higher among adolescents who had secondary education in our study Which is consistent with previous research [2,31]. Our results showed an association with lifestyle variables such as sleeping and health satisfaction (Table 5). Respondents with incomplete physical activities were more depressed than those who did their physical activities regularly, while factors such as health and sleeping satisfaction, using mobile, and living with family did not show any statistically significant associations with depression. Anjum et al. (2021) also found similar findings in their study.

## Strengths and limitations

This is the first study among all occupation adolescents in this area which we feel is one of the primary strengths of this study. The research tools used for depression and anxiety are one of its other strong points. This study has several limitations too. Firstly, this is a cross-sectional study which hampers causal attribution. The cross-sectional nature of the included studies only allowed an amalgamation of "associated factors" instead of 'risk factors. Therefore, longitudinal study is needed to clarify the causal relationship between depression and anxiety in adolescents and risk factors. Secondly, these results cannot generalize other divisions elsewhere in Bangladesh because the sample consisted of adolescents in only one division. We could not completely rule out the possibility of recall bias. Data were collected based on self-report, which is prone to information bias. Future research is needed to measure some variables that are relevant to adolescents' mental health with qualitative research. Future studies should also broaden their geographic focus beyond Mymensingh and investigate the effects of gender and culture on mental health outcomes to ensure greater generalizability. Furthermore, studies that combine biological elements with socio-environmental factors may offer a more thorough picture of Bangladeshi adolescents' mental health. Future research can help develop evidence-based policy recommendations and more successful adolescent mental health interventions by filling in these gaps.

## Policy implications and conclusions

Depression and anxiety are serious public health issues for adolescents, yet less is known about them in low- and middle-income nations like Bangladesh. This study evaluated the prevalence of anxiety and depression among 400 adolescents in Mymensingh, Bangladesh, using a cross-sectional method and validated instruments (the PHQ-9 and GAD-7). This study used ordinal logistic regression to find out how sociodemographic and lifestyle factors are associated with mental health outcomes. According to our research, there are notable associations between late adolescents' high rates of anxiety (32.8%) and depression (47.8%) with their parents' educational attainment, cell phone use, physical activity, financial stability, and lifestyle satisfaction. Mental health illnesses were more common in adolescents from rural areas, those whose fathers had no formal education, and those without family savings. Additionally, higher rates of anxiety and depression were reported by students, single adolescents, and people who were unhappy with their sleep and health.

The findings might be utilized in formulating appropriate policies for the betterment of the mental health condition of the future generations of the country. The findings of this study demonstrate the critical need for focused mental health services and early intervention for adolescents, especially those from low-income socioeconomic backgrounds. Policy interventions should concentrate on incorporating mental health education into community-based awareness campaigns and school curricula to address these issues. To improve mental health services at the primary healthcare level, the government and public health organizations should train medical professionals to screen for and assist adolescents who are depressed and anxious. Furthermore, parental education campaigns, digital well-being projects, and financial aid programs for low-income families may all help lessen the major socioeconomic stressors that contribute to teenage mental health problems.

In addition, the government ought to start implementing specific policy changes. To assist unmarried teenagers in overcoming social and emotional obstacles the government should implement policies focusing on peer support groups, counseling services, and community-based mental health education programs. Digital literacy and online well-being campaigns should teach teenagers about responsible screen time to lessen the negative effects of mobile phone use on mental health. By incorporating psychological assistance into primary healthcare and implementing telemedicine programs to guarantee accessibility, mental health services for adolescents living in rural areas should be extended. To address the impact of parental education on adolescent mental health parental awareness programs should inform families, especially those in low-literacy households about the value of emotional support. To lessen depression among businessmen, economic stress should be reduced by implementing financial assistance programs including low-interest business loans, financial literacy training, and entrepreneurship support efforts. Campaigns to raise awareness about mental health support at the workplace and make psychological counseling services more accessible should also be encouraged.

## Acknowledgments

We would like to thank the participants in this study. We would also like to thank the data collectors for their contribution.

## Author contributions

**Conceptualization:** Roni Khatun, Md. Syful Islam.

**Data curation:** Roni Khatun, Md. Syful Islam.

**Formal analysis:** Roni Khatun, Md. Syful Islam.

**Investigation:** Roni Khatun.

**Methodology:** Roni Khatun, Salma Akter Urme, Md. Syful Islam.

**Software:** Roni Khatun, Md. Syful Islam.

**Supervision:** Md. Syful Islam.

**Writing – original draft:** Roni Khatun, Salma Akter Urme, Md. Syful Islam.

**Writing – review & editing:** Roni Khatun, Salma Akter Urme, Md. Syful Islam.

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
