## [Decision Letter · Decision Letter 0]

18 Oct 2024

PONE-D-24-28645Prevalence of depression, anxiety and its association with socio-demographic factors among adolescents aged 15 to 21 in Mymensingh division, Bangladesh: Results from a cross-sectional studyPLOS ONE

Dear Dr. Islam,

Thank you for submitting your manuscript to PLOS ONE. After careful consideration, we feel that it has merit but does not fully meet PLOS ONE’s publication criteria as it currently stands. Therefore, we invite you to submit a revised version of the manuscript that addresses the points raised during the review process.

We look forward to receiving your revised manuscript.

Kind regards,

AKM Alamgir, PhD

Academic Editor

PLOS ONE

Journal Requirements:

4. We note you have included a table to which you do not refer in the text of your manuscript. Please ensure that you refer to Table 1 in your text; if accepted, production will need this reference to link the reader to the Table.

Reviewers' comments:

Reviewer's Responses to Questions

**Comments to the Author**

1. Is the manuscript technically sound, and do the data support the conclusions?

Reviewer #1: Yes

Reviewer #2: Yes

Reviewer #3: Yes

2. Has the statistical analysis been performed appropriately and rigorously? 

Reviewer #1: Yes

Reviewer #2: Yes

Reviewer #3: No

3. Have the authors made all data underlying the findings in their manuscript fully available?

Reviewer #1: Yes

Reviewer #2: No

Reviewer #3: No

4. Is the manuscript presented in an intelligible fashion and written in standard English?

Reviewer #1: Yes

Reviewer #2: Yes

Reviewer #3: Yes

5. Review Comments to the Author

Reviewer #1: Thanks for receiving the opportunity to review your paper. Overall, I appreciate the thoroughness and importance of your study on Prevalence of depression, anxiety and its association with socio-demographic factors among adolescents aged 15 to 21 in Mymensingh division, Bangladesh: Results from a cross-sectional study. However, this manuscript must undergo minor revisions before it can be considered for publication.

The title needs to be modified; most of the research shows that adolescents are between 10-19 years or mention in title late adolescents age. The title should be Prevalence of depression, anxiety, and its association with socio-demographic factors among late adolescents age in Mymensingh division, Bangladesh: A cross-sectional study.

In abstract heading, introduction needs to change into background and keywords should be in MeSH form and alphabetical order.

Introduction: This introduction provides a good overview of the relevance of depression and anxiety among adolescents in Bangladesh. However, some references can be updated or expanded to reflect more recent studies, particularly those related to global mental health trends. The focus on Bangladesh is important, but perhaps elaborating more on the global context before zooming into Bangladesh could provide better framing

In introduction section line 60-65 needs to verify According to the World Health Organization (WHO), depression and anxiety are the most common mental health disorders among late adolescents (15-24); where depressive disorders are characterized by sadness, loss of interest, feelings of guilt or low self-esteem, disturbed sleep or appetite, feelings of tiredness or poor concentration and anxiety disorders are characterized by feelings of fear, dread, and uneasiness (5).With the provided reference, especially for the age group.

Methodology: The methodology section of the manuscript provides a detailed account of the study design and data collection process. However, several aspects need clarification and improvement to enhance the scientific rigor and reproducibility of this study.

The total number of 400 subjects is appropriate, but more detail is needed about sample size calculation.

Author need to mention how many researchers were involved in the data collection.

More information about potential biases in data collection, such as how respondents were recruited, would enhance transparency.

Results: Table 1 needs to cite in text.

Discussion: The discussion does a good job of comparing your findings to existing literature. However, the implications of your findings for future policy or practice could be expanded. What specific actions should be taken by policymakers or practitioners in Bangladesh as a result of your findings? Consider expanding on the limitations of the cross-sectional design in more detail. Additionally, potential avenues for future research based on your findings could be elaborated upon.

Conclusion: The conclusion is clear but could benefit from further emphasizing the practical applications of the study. For instance, what specific steps can be taken to improve mental health outcomes based on your findings?

Reviewer #2: Dear Young Graduates,

Thank you for your valuable work on researching adolescents' mental health issues. Your effort is commendable, and I appreciate the dedication that went into this study. However, I believe some revisions are required to translate it into a publishable paper as follows:

1. Introduction: It would be beneficial to include a paragraph that articulates your motivation for this study and outlines its novel and unique contributions. This will set a strong foundation for your readers.

2. Methodology:

o In the subsection on study design, setting, and participants, you've mentioned the use of a multi-stage sampling technique. Please provide a justification for this method, along with relevant references.

o You stated that Upazilas were randomly selected in the second stage, followed by unions and villages. However, the total number of Upazilas, unions, and villages and how many of them were chosen are not mentioned. Could you elaborate on the random selection process? Additionally, why did you limit your recruitment to rural adolescents, excluding urban and suburban populations? With potentially hundreds of thousands of adolescents available, how did you arrive at the 400 participants? Clarifying whether this was based on convenience or a first-come-first-served approach was adopted.

3. Measures: The variables you've included require justification and references. Consider discussing why other relevant variables, such as the family history of depression or anxiety, educational outcomes, financial stress, household wealth, physical health indicators (e.g., BMI), drug addiction, frustration about life or love, etc. were not measured.

4. Statistical Methods: Your paper currently lacks detail on the statistical methods used. Please include a discussion of ordinal logistic regression, highlighting its appropriateness for this study compared to other methods like OLS, multinomial logit, or probit, etc.

5. Conceptual Framework: Instead of presenting your conceptual framework in the Statistical Analysis section, please create a dedicated section after the introduction. Moreover, Figure 1 indicates that socio-economic factors have both direct and indirect effects on depression and anxiety via lifestyle factors. If this is the case, you should illustrate how the indirect effects occur through your ordinal logistic regression using instrumental variables. If you wish to simplify the model due to insufficient instrumental variables in your dataset, consider modifying the figure to show only direct effects.

6. Language Sensitivity: In your script, terms like “No education” and “father/mother had no education” may come off as imprudent. Consider using “without formal education” and father/mother without formal education” instead for a more respectful tone.

7. Results Section: On pages 11-12, you mention that “The ratio of moderate to severe depression was higher in (i) males vs. females, (ii) older adolescents (19-21 years) vs. years age groups, … , with severe anxiety (Table 2)” which are meaningless. Similarly, on pages 14-15, mentions that “The proportion of moderate to severe anxiety was higher in (i) males vs. females, (ii) older adolescents (19-21 years) vs. 15-16 years age groups, (iii) Muslim vs. non-Muslim…..” Please rewrite these findings in a more digestible manner.

8. Specifications or robustness tests: Please include specifications or robustness tests for your ordinal logistic regression results to validate your findings.

9. Discussion: Instead of solely comparing your findings to other studies, begin each paragraph in the discussion section by explaining your own results on relevant variables before making comparisons.

10. Policy implications, limitations of the study, and suggestions for future research: Your study lacks discussion on policy implications, limitations of the study, and suggestions for future research. Add them in the conclusion section in detailed paragraphs.

Remember, research is an ongoing learning journey. I hope these suggestions aid your growth and refinement in academia. I look forward to seeing your revised manuscript soon!

Reviewer #3: The authors have done commendable work in addressing a crucial and often under-researched issue of adolescent mental health in Bangladesh, providing valuable insights into the prevalence and socio-demographic factors associated with depression and anxiety. To further strengthen the manuscript, the following suggestions are offered: enhancing the clarity and structure of the abstract by including specific numerical findings and organizing it into subsections; refining the introduction to focus more on the local context and research gaps; providing a more detailed description of the sampling strategy, inclusion criteria, and validation of tools in the methods section; streamlining the presentation of results with a focus on significant findings, and incorporating stratified analyses; deepening the discussion with a critical comparison to existing literature and a more thorough exploration of implications for policy; and finally, including measures of effect size, interaction terms, and model fit diagnostics in the statistical analysis

1. Abstract

o The abstract lacks structure and detailed quantitative results. Although it mentions the prevalence of depression and anxiety, it does not provide sufficient breakdown of findings or statistical measures.

o The objectives are stated generally, and the results do not clearly specify the key predictors or effect sizes.

• Recommendations:

o Structure: Use subheadings like "Background," "Methods," "Results," and "Conclusions" for better clarity.

o Be Specific About Prevalence: Instead of saying, “Almost half (47.8%) of adolescents experience depression and over a third (32.8%) experience anxiety,” specify the severity levels, e.g., "47.8% had depression (---% moderate, ---% moderately severe, ---% severe), and 32.8% had anxiety (---% moderate, ---% severe)."

o Include Key Predictors: Mention the main predictors found in the regression analysis. For instance, "Depression was significantly associated with father's education level (p < 0.05) and lack of regular physical activity (p < 0.01)."

o Quantify the Associations: Report odds ratios (ORs) for key associations. For example, "Adolescents whose fathers had no education were --- times more likely to experience depression (95% CI: ---–---)."

2. Introduction

o The introduction is broad and spends too much time on global mental health statistics without quickly narrowing down to the local context in Bangladesh.

o The research objectives are somewhat buried and not explicitly detailed.

• Recommendations:

o Focus on Local Context Early On: Begin by stating the lack of adolescent mental health research in Bangladesh, specifically in rural areas and among older adolescents (19-21). Use national statistics from the 2019 Bangladesh National Mental Health Survey early in the introduction.

o Define the Research Gap Clearly: Emphasize the gaps that this study addresses, such as limited research on out-of-school adolescents or those living in rural areas.

o Explicit Research Objectives: State the objectives clearly. For example: "This study aims to (1) estimate the prevalence of depression and anxiety among adolescents aged 15-21 in Mymensingh, and (2) identify socio-demographic factors associated with these mental health conditions, including parental education, physical activity, and rural versus urban residence."

3. Methods

o The sampling process and criteria for inclusion/exclusion are not adequately detailed, making it difficult to assess the study's generalizability.

o There is insufficient explanation of the tools used (PHQ-9 and GAD-7) and how they were validated for use in the local context.

• Recommendations:

o Clarify Sampling Strategy: Expand on the multi-stage sampling technique. For example, describe the process for selecting districts, Upazilas, and villages, and how many participants were recruited from each stage. Specify if any steps were taken to ensure diversity in the sample.

o Explain Inclusion/Exclusion Criteria Clearly: Mention if adolescents with ongoing mental health treatments or specific conditions were excluded. Clarify how participants were approached, especially if they were out-of-school.

o Validation of Tools: Explain the reliability and validity of the PHQ-9 and GAD-7 scales in the local context. If these tools were used in other Bangladeshi studies, provide citations. Include any cultural adaptations made to the tools.

o Data Availability: Mention where the data can be accessed post-publication (according to PLOS ONE requirements) or explain the ethical restrictions.

4. Results

o The presentation of results is somewhat dense and difficult to follow, especially with large tables containing many variables. Non-significant variables are listed without further discussion.

o There is minimal stratification of results by key factors like gender, rural/urban residence, or age.

• Recommendations:

o Streamline Tables: Focus on significant results in the main text and relegate non-significant findings to supplementary materials. For example, condense Table 2 by only presenting variables with p-values < 0.05 and showing their odds ratios.

o Include More Stratified Analyses: Break down the prevalence of depression and anxiety by gender and location (rural vs. urban). For example, report if males had higher depression rates in rural areas than females.

o Graphical Representation: Add visual elements like bar charts or forest plots to make the results more accessible. For example, display odds ratios with confidence intervals for key predictors.

5. Discussion

o The discussion is more of a summary than a critical analysis. It does not adequately engage with existing literature to explain why some findings may differ from other studies.

o There is minimal exploration of the implications of the findings for policy and practice in Bangladesh.

• Recommendations:

o Engage Critically with Other Studies: Compare findings with previous studies in Bangladesh and nearby regions. For example, discuss why the current study found higher depression rates in males, while other studies reported higher rates in females. Suggest possible cultural, social, or methodological reasons for these differences.

o Implications for Policy: Go beyond suggesting that mental health services need improvement. Provide specific recommendations, such as implementing school-based mental health programs or integrating mental health education in rural communities.

o Explain Limitations Thoroughly: The current limitations section is brief. Discuss potential biases (e.g., self-report bias), limitations due to the cross-sectional design, and any sampling issues that may affect generalizability. For example, mention the overrepresentation of rural adolescents and how it might skew the findings.

o Future Research Suggestions: Recommend longitudinal studies or qualitative research to explore causality and understand the lived experiences of adolescents with mental health issues.

6. Conclusion

o The conclusion restates the findings without providing a strong summary of their implications or suggesting next steps.

• Recommendations:

o Summarize Key Findings Clearly: For example, "This study found high prevalence rates of depression and anxiety, particularly among rural adolescents and those with low parental education, indicating an urgent need for targeted interventions."

o Convey Policy Recommendations Briefly: Instead of generic suggestions, be specific, e.g., "Policy efforts should focus on training healthcare workers in rural areas to identify and manage adolescent mental health issues."

o Avoid Repetition: Do not repeat points already discussed in detail in the results or discussion. Instead, provide a concise statement on the significance of the findings.

7. Tables and Figures

o Some tables are too detailed, making them difficult to interpret quickly. There is no use of figures to illustrate key findings.

• Recommendations:

o Condense Tables: For example, in Table 2, highlight only significant results in the main manuscript. Move less relevant details to supplementary materials.

o Add Figures: Consider using a forest plot to illustrate the odds ratios and confidence intervals from the logistic regression. Add bar charts for the prevalence of depression and anxiety across different age groups or by gender.

8. Statistical Analysis

o The analysis could benefit from including measures of effect size and considering interactions between key variables.

• Recommendations:

o Include Effect Sizes (e.g., Cramér's V for Chi-Square Tests): This would add context to the significance tests.

o Test the Proportional Odds Assumption: For ordinal logistic regression, confirm that this assumption holds. If not, consider using alternative models.

o Include Interaction Terms: Explore interactions, for example, between gender and rural residence, to see if the effects of socio-demographic factors on mental health outcomes differ across subgroups.

o Model Fit Diagnostics: Report Pseudo R-squared values or other model fit statistics for the ordinal logistic regression.

6. PLOS authors have the option to publish the peer review history of their article (what does this mean? ). If published, this will include your full peer review and any attached files.

**Do you want your identity to be public for this peer review?** For information about this choice, including consent withdrawal, please see our Privacy Policy .

Reviewer #1: No

Reviewer #2: No

Reviewer #3: **Yes: ** Adrija Roy

---

## [Author Response · Author response to Decision Letter 1]

11 Dec 2024

Response to Journals Requirements:

SL Reviewers’ comment Author's response

1 Please ensure that your manuscript meets PLOS ONE's style requirements, including those for file naming.

Author's response: Thank you for your comment we checked PLOS ONE’s style requirements and addressed them accordingly.

2 You indicated that you had ethical approval for your study. In your Methods section, please ensure you have also stated whether you obtained consent from parents or guardians of the minors included in the study or whether the research ethics committee or IRB specifically waived the need for their consent.

Author's response: Thank you for raising this point. We unintentionally missed this point, but we obtained consent from the parents or guardians of the respondents aged below 18. We’ve now included this statement in the ethical consideration section under methodology.

3 We note that you have indicated that there are restrictions to data sharing for this study. For studies involving human research participant data or other sensitive data, we encourage authors to share de-identified or anonymized data. However, when data cannot be publicly shared for ethical reasons, we allow authors to make their data sets available upon request.

Author's response: Thank you for raising this issue. While obtaining ethical review and collecting data we didn’t obtain permission for data sharing from the respondents. However, for the betterment of the scientific community, we are okay to share data upon request.

4 We note you have included a table to which you do not refer in the text of your manuscript. Please ensure that you refer to Table 1 in your text; if accepted, production will need this reference to link the reader to the Table.

Author's response: Thank you, we cited Table 1 in the text.

Response to reviewers’ comments:

Reviewer #1:

1 Thanks for receiving the opportunity to review your paper. Overall, I appreciate the thoroughness and importance of your study on Prevalence of depression, anxiety and its association with socio-demographic factors among adolescents aged 15 to 21 in Mymensingh division, Bangladesh: Results from a cross-sectional study. However, this manuscript must undergo minor revisions before it can be considered for publication.

The title needs to be modified; most of the research shows that adolescents are between 10-19 years or mention in title late adolescents age. The title should be Prevalence of depression, anxiety, and its association with socio-demographic factors among late adolescents age in Mymensingh division, Bangladesh: A cross-sectional study.

Author's response: Thank you for your valuable comment and guidance. We’ve made changes in the title now it read

Prevalence of depression and anxiety and their association with socio-demographic factors among late adolescents aged 15 to 21 in Mymensingh division, Bangladesh: A cross-sectional study

2 In abstract heading, introduction needs to change into background and keywords should be in MeSH form and alphabetical order.

Author's response: We’ve addressed this comment and made changes accordingly.

3 Introduction: This introduction provides a good overview of the relevance of depression and anxiety among adolescents in Bangladesh. However, some references can be updated or expanded to reflect more recent studies, particularly those related to global mental health trends. The focus on Bangladesh is important, but perhaps elaborating more on the global context before zooming into Bangladesh could provide better framing

In introduction section line 60-65 needs to verify According to the World Health Organization (WHO), depression and anxiety are the most common mental health disorders among late adolescents (15-24); where depressive disorders are characterized by sadness, loss of interest, feelings of guilt or low self-esteem, disturbed sleep or appetite, feelings of tiredness or poor concentration and anxiety disorders are characterized by feelings of fear, dread, and uneasiness (5).With the provided reference, especially for the age group. Author's response: Thank you for your valuable comment we have rewritten this section accordingly.

4 Methodology: The methodology section of the manuscript provides a detailed account of the study design and data collection process. However, several aspects need clarification and improvement to enhance the scientific rigor and reproducibility of this study.

The total number of 400 subjects is appropriate, but more detail is needed about sample size calculation.

Author's response: Thank you for your valuable comment we have now included the sample size calculation in detail.

5 Author need to mention how many researchers were involved in the data collection.

Author's response: Thank you. We have included this in the methodology section under the sub-heading “Study design, setting, and participants”

6 More information about potential biases in data collection, such as how respondents were recruited, would enhance transparency.

Author's response: We have included this in the methodology section under the sub-heading “Study design, setting, and participants”

7 Results: Table 1 needs to cite in text.

Author's response: Thank you for your comment we’ve cited this in the text.

8 Discussion: The discussion does a good job of comparing your findings to existing literature. However, the implications of your findings for future policy or practice could be expanded. What specific actions should be taken by policymakers or practitioners in Bangladesh as a result of your findings? Consider expanding on the limitations of the cross-sectional design in more detail. Additionally, potential avenues for future research based on your findings could be elaborated upon.

Author's response: Thank you for bringing this up. We have added the implications of the findings for policy and practice in Bangladesh and some limitations in the discussion part.

9 Conclusion: The conclusion is clear but could benefit from further emphasizing the practical applications of the study. For instance, what specific steps can be taken to improve mental health outcomes based on your findings?

Author's response: Thank you for your nice comment. We have now rewritten the conclusion to address your comment.

Reviewer #2:

Overall Comment: Thank you for your valuable work on researching adolescents' mental health issues. Your effort is commendable, and I appreciate the dedication that went into this study. However, I believe some revisions are required to translate it into a publishable paper as follows:

Author's response: We appreciate your nice words. Thank you for your appreciation.

1 Introduction: It would be beneficial to include a paragraph that articulates your motivation for this study and outlines its novel and unique contributions. This will set a strong foundation for your readers.

Author's response: Thank you for your valuable comment. We’ve outlined the novelty and unique contributions in the introduction. We couldn’t add the motivation for this study in a different paragraph, but we believe the introduction section has already addressed the motivation academically.

2 Methodology:

- In the subsection on study design, setting, and participants, you've mentioned the use of a multi-stage sampling technique. Please provide a justification for this method, along with relevant references.

- You stated that Upazilas were randomly selected in the second stage, followed by unions and villages. However, the total number of Upazilas, unions, and villages and how many of them were chosen are not mentioned. Could you elaborate on the random selection process? Additionally, why did you limit your recruitment to rural adolescents, excluding urban and suburban populations? With potentially hundreds of thousands of adolescents available, how did you arrive at the 400 participants? Clarifying whether this was based on convenience or a first-come-first-served approach was adopted.

Author's response: In response to your comment, we have revised this section and added the details procedures of the multi-stage sampling technique with its justification. We have included the total number of Upazilas and unions we’ve selected. For adolescent recruitment, we’ve recruited adolescents from both rural and urban areas of these selected upazilas. We also added the details sampling procedures in the methodology section.

3 Measures: The variables you've included require justification and references. Consider discussing why other relevant variables, such as the family history of depression or anxiety, educational outcomes, financial stress, household wealth, physical health indicators (e.g., BMI), drug addiction, frustration about life or love, etc. were not measured. Author's response: Thank you for your valuable comment we have included justification and references for used variables in the measure’s subsection of the methodology section. We’ve discussed the limitation of not including these variables and mentioned them in the conclusion part for further research implications.

4 Statistical Methods: Your paper currently lacks detail on the statistical methods used. Please include a discussion of ordinal logistic regression, highlighting its appropriateness for this study compared to other methods like OLS, multinomial logit, or probit, etc.

Author's response: Thank you for your valuable comment. We have detailed the statistical methods including a discussion of ordinal logistic regression. We also compare other methods to highlight the appropriateness of using ordinal logistic regression.

5 Conceptual Framework: Instead of presenting your conceptual framework in the Statistical Analysis section, please create a dedicated section after the introduction. Moreover, Figure 1 indicates that socio-economic factors have both direct and indirect effects on depression and anxiety via lifestyle factors. If this is the case, you should illustrate how the indirect effects occur through your ordinal logistic regression using instrumental variables. If you wish to simplify the model due to insufficient instrumental variables in your dataset, consider modifying the figure to show only direct effects.

Author's response: Thank you so much for your comment we’ve placed the conceptual framework after the introduction. We also develop a simplified model due to insufficient instrumental variables showing only direct effects.

6 Language Sensitivity: In your script, terms like “No education” and “father/mother had no education” may come off as imprudent. Consider using “without formal education” and father/mother without formal education” instead for a more respectful tone.

Author's response: Thank you for your valuable observation. We’ve addressed these accordingly.

7 Results Section: On pages 11-12, you mention that “The ratio of moderate to severe depression was higher in (i) males vs. females, (ii) older adolescents (19-21 years) vs. years age groups, … , with severe anxiety (Table 2)” which are meaningless. Similarly, on pages 14-15, mentions that “The proportion of moderate to severe anxiety was higher in (i) males vs. females, (ii) older adolescents (19-21 years) vs. 15-16 years age groups, (iii) Muslim vs. non-Muslim…..” Please rewrite these findings in a more digestible manner.

Author's response: Thank you for your nice comment. We have now revised the relevant sections to address your comment, and we believe it now reads better.

8 Specifications or robustness tests: Please include specifications or robustness tests for your ordinal logistic regression results to validate your findings.

Author's response: Thank you for your suggestion. We have included robustness tests in the manuscript.

9 Discussion: Instead of solely comparing your findings to other studies, begin each paragraph in the discussion section by explaining your own results on relevant variables before making comparisons.

Author's response: Thank you for your valuable comment. We’ve changed the discussion to address this comment.

10 Policy implications, limitations of the study, and suggestions for future research: Your study lacks discussion on policy implications, limitations of the study, and suggestions for future research. Add them in the conclusion section in detailed paragraphs.

Author's response: We appreciate your comment and addressed these appropriately.

Reviewer #3:

Overall Comment

The authors have done commendable work in addressing a crucial and often under-researched issue of adolescent mental health in Bangladesh, providing valuable insights into the prevalence and socio-demographic factors associated with depression and anxiety. To further strengthen the manuscript, the following suggestions are offered: enhancing the clarity and structure of the abstract by including specific numerical findings and organizing it into subsections; refining the introduction to focus more on the local context and research gaps; providing a more detailed description of the sampling strategy, inclusion criteria, and validation of tools in the methods section; streamlining the presentation of results with a focus on significant findings, and incorporating stratified analyses; deepening the discussion with a critical comparison to existing literature and a more thorough exploration of implications for policy; and finally, including measures of effect size, interaction terms, and model fit diagnostics in the statistical analysis.

Author's response: Thank you for your insights, appreciation, and valuable comments. We have addressed your comments and the manuscript now reads better.

1 Abstract

1.1 The abstract lacks structure and detailed quantitative results. Although it mentions the prevalence of depression and anxiety, it does not provide sufficient breakdown of findings or statistical measures.

Author's response: Thank you for your valuable comment, we have addressed this accordingly.

1.2 The objectives are stated generally, and the results do not clearly specify the key predictors or effect sizes.

Author's response: Thank you for your valuable comment, we have addressed this accordingly.

1.3 Structure: Use subheadings like "Background," "Methods," "Results," and "Conclusions" for better clarity.

Author's response: Thank you for your recommendations, we’ve changed accordingly.

1.4 Be Specific About Prevalence: Instead of saying, “Almost half (47.8%) of adolescents experience depression and over a third (32.8%) experience anxiety,” specify the severity levels, e.g., "47.8% had depression (---% moderate, ---% moderately severe, ---% severe), and 32.8% had anxiety (---% moderate, ---% severe)."

Author's response: Thank you for your nice comment. We have now revised the relevant sections to address your comment, and we believe it now reads better. This now reads-

The overall prevalence of depression was 47.8% (33.5% moderate, 10.5% moderately severe, and 3.8% severe depression), and anxiety 32.8% (31% moderate and 1.8% severe anxiety). Across most of the socio-demographic and lifestyle-rel

---

## [Decision Letter · Decision Letter 1]

29 Dec 2024

PONE-D-24-28645R1Prevalence and socio-demographic correlates of depression and anxiety among late adolescents (15 to 21 years) in Mymensingh division, Bangladesh: A cross-sectional studyPLOS ONE

Dear Dr. Islam,

Thank you for submitting your manuscript to PLOS ONE. After careful consideration, we feel that it has merit but does not fully meet PLOS ONE’s publication criteria as it currently stands. Therefore, we invite you to submit a revised version of the manuscript that addresses the points raised during the review process.

Please check the comments of the learned reviewer (Amit Roy). Reply those comments and fix the text accordingly. 

We look forward to receiving your revised manuscript.

Kind regards,

AKM Alamgir, PhD

Academic Editor

PLOS ONE

Journal Requirements:

Reviewers' comments:

Reviewer's Responses to Questions

**Comments to the Author**

1. If the authors have adequately addressed your comments raised in a previous round of review and you feel that this manuscript is now acceptable for publication, you may indicate that here to bypass the “Comments to the Author” section, enter your conflict of interest statement in the “Confidential to Editor” section, and submit your "Accept" recommendation.

Reviewer #1: All comments have been addressed

Reviewer #2: (No Response)

Reviewer #3: All comments have been addressed

2. Is the manuscript technically sound, and do the data support the conclusions?

Reviewer #1: Yes

Reviewer #2: Yes

Reviewer #3: Yes

3. Has the statistical analysis been performed appropriately and rigorously? 

Reviewer #1: Yes

Reviewer #2: No

Reviewer #3: Yes

4. Have the authors made all data underlying the findings in their manuscript fully available?

Reviewer #1: No

Reviewer #2: No

Reviewer #3: No

5. Is the manuscript presented in an intelligible fashion and written in standard English?

Reviewer #1: Yes

Reviewer #2: No

Reviewer #3: Yes

6. Review Comments to the Author

Reviewer #1: Thank you for the opportunity to review your research paper. I wish you further success and look forward to seeing your work in print.

Reviewer #2: Dear Authors,

Thank you for your revision. However, it is not yet adequate and complete. Please revise your manuscript further, addressing the following suggestions:

1. In the abstract, you use different p-value specifications (e.g., OR = 0.15; 95% CI = 0.03-0.75, p = 0.021 or OR = 0.06; 95% CI = 0.01-0.22, p < 0.000). The p-value format should be consistent. Please use a uniform presentation, such as "<0.05" instead of "p = 0.021," throughout the abstract.

2. I asked you to add a new section after the introduction for a Conceptual Framework and to discuss it in detail, explaining how each variable in the diagram relates to the prevalence of depression and anxiety. Instead, you added a brief summary. Please address this issue by providing a detailed, thoughtful discussion in 3-6 paragraphs in a separate Conceptual Framework section after introduction.

3.

4. In the Methodology section, I requested an explanation of the random selection process of participants. However, you did not provide details on the methodology used, such as whether a lottery or random number generation was employed for random selection at each stage of sampling. Furthermore, I raised the question of how you choose 400 participants when there are potentially hundreds of thousands of adolescents available. There was no response regarding this crucial issue. Please explain in detail how the 400 participants were selected from the millions of adolescents in the study area, ideally in a separate paragraph. If randomly selected, which database you used and how did you access it, if not which other method.

5. I recommend that you justify and reference the variables you included in the study. “Please discuss why other relevant variables—such as family history of depression or anxiety, educational outcomes, financial stress, household wealth, physical health indicators (e.g., BMI), drug addiction, frustration about life or love, etc.—were not measured.” In your response, you stated that justification and references were provided in the methodology section. However, I found no such discussion in the revised manuscript. You need to clarify why the variables you selected were chosen and explain why others were excluded, citing relevant studies or references.

6. Regarding statistical methods, your paper currently lacks detail on the methods used. You mentioned that ordinal logistic regression was discussed and compared with other methods to highlight its appropriateness for the study, but I could not find any detailed explanation of this in the revised manuscript. Please include a statistical methods section with a discussion of ordinal logistic regression, including its equation and parameters.

7. In response to my comment 8, you mentioned that "robustness tests have been included in the manuscript." However, I could not find any robustness tests or results in the revised manuscript. Additionally, you have 23 variables in the study, but only 8-9 variables are included in the ordinal logistic regression, which suggests that your model may be manipulated. To avoid bias, you need to incorporate all explanatory variables mentioned in the descriptive statistics and run the ordinal logistic regression with these variables included. Also, variables such as age, gender, and family income are crucial. Why were they excluded from the regression? These variables need to be included, and the regression should be rerun to ensure accurate results.

8. In the discussion section, you included a lot of information from other research but failed to properly discuss your own findings. You need to first compare your findings with those from other studies, and then explain how the local context of your study area may account for differences from other research. This comparison has not been done adequately.

9. In comment 10, I asked you to add a discussion on policy implications, study limitations, and suggestions for future research in the conclusion section. Unfortunately, no improvement has been made in this regard. Please rewrite the conclusion in 3-5 detailed paragraphs: one sentence on background and methodology, one sentence on results and findings, one on policy suggestions based on your findings, one on limitations, and one on future research. This should be done thoroughly.

10. Your finding that "unmarried adolescents, mobile phone users, rural adolescents, and those whose fathers and mothers were without formal education were more likely to experience moderate to severe depression and anxiety than their counterparts" should be reflected in policy suggestions. In separate sentences, provide specific policies, programs, or actions that can address each of these variables and help improve mental health outcomes.

11. In the abstract, you mentioned that "depression was more likely to be associated with those who were businessmen and day laborers." This should be analyzed in the results section. You need to explore why these groups are vulnerable to depression and suggest policies to reduce mental stress among these working groups.

12. The last paragraph you added in the discussion should be rewritten in clearer, more readable language and placed in the limitations paragraph of the conclusion.

13. In my previous comments 7 and 8, I requested clarification and made suggestions to improve readability. However, the revised manuscript still contains sections that are not audience-friendly or easy to read. Please revise these paragraphs to improve clarity and coherence.

I look forward to receiving the revised manuscript.

Reviewer #3: The manuscript effectively addresses a critical and under-researched issue of adolescent mental health in Bangladesh, providing valuable insights into the prevalence of depression and anxiety and their socio-demographic correlates. The authors have made significant revisions based on reviewer feedback, improving the structure, clarity, and scientific rigor of the study. Key strengths include the detailed statistical analysis with effect sizes, the inclusion of specific policy recommendations, and a critical comparison with existing literature. However, a few minor improvements can further enhance the manuscript's quality. These include explicitly clarifying whether the proportional odds assumption was tested, and expanding on the implementation challenges of proposed policies in the Bangladeshi context. Overall, the manuscript is well-prepared and, with these refinements, is suitable for publication, contributing meaningfully to the global conversation on adolescent mental health.

7. PLOS authors have the option to publish the peer review history of their article (what does this mean? ). If published, this will include your full peer review and any attached files.

**Do you want your identity to be public for this peer review?** For information about this choice, including consent withdrawal, please see our Privacy Policy .

Reviewer #1: No

Reviewer #2: No

Reviewer #3: **Yes: ** Dr. Adrija Roy

---

## [Author Response · Author response to Decision Letter 2]

14 Feb 2025

Response to Journal Requirements:

Journal Requirements:

Please review your reference list to ensure that it is complete and correct. If you have cited papers that have been retracted, please include the rationale for doing so in the manuscript text or remove these references and replace them with relevant current references. Any changes to the reference list should be mentioned in the rebuttal letter that accompanies your revised manuscript. If you need to cite a retracted article, indicate the article’s retracted status in the References list and also include a citation and full reference for the retraction notice.

Author's response:

I appreciate your input. we have thoroughly examined our list of references to make sure it is accurate and comprehensive. We’ve made substantial changes in the reference list of the manuscript. Our citations contained no retracted articles.

Response to reviewers’ comments:

Reviewer #1:

Reviewers’ comment: Thank you for the opportunity to review your research paper. I wish you further success and look forward to seeing your work in print.

Author's response: Thank you for your good wishes and for helping us improve the manuscript with your insightful feedback.

Reviewer #2:

Reviewers’ comment: Thank you for your revision. However, it is not yet adequate and complete. Please revise your manuscript further, addressing the following suggestions.

Author's response: Thank you for your comment. We’ve carefully addressed all your comments. Hope this version will be okay to answer your queries. We’re thankful for your constructive comments and feedback. These really help to improve the quality of our manuscript.

Reviewers’ comment 01: In the abstract, you use different p-value specifications (e.g., OR = 0.15; 95% CI = 0.03-0.75, p = 0.021 or OR = 0.06; 95% CI = 0.01-0.22, p < 0.000). The p-value format should be consistent. Please use a uniform presentation, such as "<0.05" instead of "p = 0.021," throughout the abstract.

Author's response: Thank you for your valuable comment and guidance. We have rewritten the abstract section following your comment.

Reviewers’ comment 02: I asked you to add a new section after the introduction for a Conceptual Framework and to discuss it in detail, explaining how each variable in the diagram relates to the prevalence of depression and anxiety. Instead, you added a brief summary. Please address this issue by providing a detailed, thoughtful discussion in 3-6 paragraphs in a separate Conceptual Framework section after introduction.

Author's response: Thank you for your comment. We have addressed your comment and included a new section at the end of the introduction.

Reviewers’ comment 03: In the Methodology section, I requested an explanation of the random selection process of participants. However, you did not provide details on the methodology used, such as whether a lottery or random number generation was employed for random selection at each stage of sampling. Furthermore, I raised the question of how you choose 400 participants when there are potentially hundreds of thousands of adolescents available. There was no response regarding this crucial issue. Please explain in detail how the 400 participants were selected from the millions of adolescents in the study area, ideally in a separate paragraph. If randomly selected, which database you used and how did you access it, if not which other method.

Author's response: Thank you for your detail comment. In our study, we randomly selected every unit but at the final stage due to lack of list or database of adolescents of the data collection area we had to select respondents conveniently following inclusion criteria.

Reviewers’ comment 04: I recommend that you justify and reference the variables you included in the study. “Please discuss why other relevant variables—such as family history of depression or anxiety, educational outcomes, financial stress, household wealth, physical health indicators (e.g., BMI), drug addiction, frustration about life or love, etc.—were not measured.” In your response, you stated that justification and references were provided in the methodology section. However, I found no such discussion in the revised manuscript. You need to clarify why the variables you selected were chosen and explain why others were excluded, citing relevant studies or references.

Author's response: Thank you for your valuable comment. The justification for variable selection has now been specifically addressed in the Methodology section. The elimination of some variables due to issues with measurement, data dependability, and ethics has been supported by further references. In the Discussion section, we identify these shortcomings and recommend more research in these areas. This reads,

Adolescent mental health is influenced by several factors, including a family history of depression or anxiety, educational outcomes, financial stress, household wealth, physical health indicators (e.g., BMI), drug addiction, frustration about life or love, etc. However, these factors were excluded for the following reasons:

Family history of depression or anxiety: Due to stigma and lack of official diagnoses, family histories of anxiety and depression are frequently underreported in Bangladesh (21,23,30). Furthermore, it was thought to be difficult for teenagers to accurately self-report on the mental health of their parents.

Educational outcomes: Although academic stress may play a role, our research concentrated on more general sociodemographic factors. Parental educational attainment was used to measure the indirect impact of education.

Financial stress, and wealth: We employed monthly family income, spending, and savings as a proxy for economic status in place of explicit financial stress measures, as these have been often employed in related research (12).

Physical health indicators (e.g., BMI): Physical health indicators e.g., BMI were excluded from data collection due to practical limitations in measuring anthropometric data in field-based surveys. However, we employed self-reported health satisfaction as a proxy following similar studies (19).

Drug addiction: We avoided difficult items that could result in underreporting because of ethical considerations and the self-reported nature of our study. Nonetheless, one relevant lifestyle component was smoking status.

Frustration about Life or Love: Although psychological stressors are significant, our study concentrated on behavioral and sociodemographic factors that have been validated in prior research and can be quantified objectively.

Reviewers’ comment 05: Regarding statistical methods, your paper currently lacks detail on the methods used. You mentioned that ordinal logistic regression was discussed and compared with other methods to highlight its appropriateness for the study, but I could not find any detailed explanation of this in the revised manuscript. Please include a statistical methods section with a discussion of ordinal logistic regression, including its equation and parameters.

Author's response: Thank you for your detailed feedback. We’ve made changes in the method section to address your comment. We have included a statistical methods section with a discussion of ordinal logistic regression, including its equation and parameters. The statistical analysis section is now read-

Data entry, editing, and analysis were performed using the Statistical Package for the Social Sciences (SPSS) software program, version 26.0. We used descriptive statistics (e.g., frequencies, percentages, mean, standard deviation) and first-order analysis (i.e., chi-square tests). Variables that had a p-value of less than or equal to 0.05 were further analysed using ordinal logistic regression. Although multinomial logit and ordinal logistic regression are used for categorical data with more than two categories. The association of variables was considered statistically significant if the two-sided p-value was less than .05.

When the dependent variable of our study, depression and anxiety level, were is measured in ordinal scale (Minimal, mild, moderate, moderately severe, and severe) ordered categories so, we employed ordinal logistic regression (OLR). is best whereas when the dependent variable is categorical but has no link between the categories then should use multinomial logit. So, we think our data have ordered categories (Minimal, mild, moderate, moderately severe, and severe) and fit the ordinal logistic regression. OLR is suitable when the outcome variable is categorically ordered and allows us to estimate the probability of falling into a higher category of anxiety or depression while a few predictor variables are accounted.

Ordinal Logistic Regression Model:

Based on the cumulative odds technique, the ordinal logistic regression model makes the proportional odds assumption about the link between independent factors and the ordinal response variable. The following is a representation of the model:

Log(P(Y≤j)/P(Y>j) = β0+β1X1+β2X2+⋯+BkXk

Here,

P (Y≤j) is the cumulative probability of the outcome variable being at or below category j,

P(Y>j) is the probability of being above category j,

β0 is the intercept,

X1, X2, ..., Xk are the independent variables (e.g., age, gender, parental education, family income, physical activity, smoking),

β1, β2, ..., βk are the regression coefficients representing the change in the log-odds of being in a higher category for a one-unit increase in the predictor variable.

Reviewers’ comment 06: In response to my comment 8, you mentioned that "robustness tests have been included in the manuscript." However, I could not find any robustness tests or results in the revised manuscript. Additionally, you have 23 variables in the study, but only 8-9 variables are included in the ordinal logistic regression, which suggests that your model may be manipulated. To avoid bias, you need to incorporate all explanatory variables mentioned in the descriptive statistics and run the ordinal logistic regression with these variables included. Also, variables such as age, gender, and family income are crucial. Why were they excluded from the regression? These variables need to be included, and the regression should be rerun to ensure accurate results.

Author's response: Thank you for your comment, and we’re very sorry that we mistakenly forgot to add this. Now we’ve added robustness tests in methodology section of the manuscript. Following best practices in public health research, we first reduced over fitting and multicollinearity by including only variables with p < 0.05 from the bi variate analysis in the ordinal logistic regression (Jeelani et al., 2022; Magklara et al., 2015b; Mohamad et al., 2021). We do, however, recognize the worry about possible bias resulting from variable exclusion. The consistency of the results confirmed the strength of our conclusions. To improve interpretability and prevent statistical noise, we stuck with the stepwise strategy, making sure that only variables that had been shown to be significant in bi variate analysis were included in the regression.

Reviewers’ comment 07: In the discussion section, you included a lot of information from other research but failed to properly discuss your own findings. You need to first compare your findings with those from other studies and then explain how the local context of your study area may account for differences from another research. This comparison has not been done adequately.

Author's response: Thank you for your comment. We have revised the discussion to address your comment.

Reviewers’ comment 08: In comment 10, I asked you to add a discussion on policy implications, study limitations, and suggestions for future research in the conclusion section. Unfortunately, no improvement has been made in this regard. Please rewrite the conclusion in 3-5 detailed paragraphs: one sentence on background and methodology, one sentence on results and findings, one on policy suggestions based on your findings, one on limitations, and one on future research. This should be done thoroughly.

Author's response: We appreciate your thoughtful comments, and we have fully rewritten the conclusion section to address your comment.

Reviewers’ comment 09: You’re finding that "unmarried adolescents, mobile phone users, rural adolescents, and those whose fathers and mothers were without formal education were more likely to experience moderate to severe depression and anxiety than their counterparts" should be reflected in policy suggestions. In separate sentences, provide specific policies, programs, or actions that can address each of these variables and help improve mental health outcomes.

Author's response: Thank you for your comment. We have address this in the policy recommendations.

Reviewers’ comment 10: In the abstract, you mentioned that "depression was more likely to be associated with those who were businessmen and day laborers." This should be analyzed in the results section. You need to explore why these groups are vulnerable to depression and suggest policies to reduce mental stress among these working groups.

Author's response: Thank you for your insight. We have already analyzed businessmen and day labourer in the result section. Why these groups are vulnerable to depression are included in discussion part and policy suggestion for this groups have added accordingly.

Reviewers’ comment 11: The last paragraph you added in the discussion should be rewritten in clearer, more readable language and placed in the limitations paragraph of the conclusion.

Author's response: Thank you and we have rewritten the whole conclusion part.

Reviewers’ comment 12: In my previous comments 7 and 8, I requested clarification and made suggestions to improve readability. However, the revised manuscript still contains sections that are not audience-friendly or easy to read. Please revise these paragraphs to improve clarity and coherence.

Author's response: Thank you for your suggestions. We have addressed this comment and improve the manuscript to make it audience friendly.

Reviewer #3:

Reviewers’ comment: The manuscript effectively addresses a critical and under-researched issue of adolescent mental health in Bangladesh, providing valuable insights into the prevalence of depression and anxiety and their socio-demographic correlates. The authors have made significant revisions based on reviewer feedback, improving the structure, clarity, and scientific rigor of the study. Key strengths include the detailed statistical analysis with effect sizes, the inclusion of specific policy recommendations, and a critical comparison with existing literature. However, a few minor improvements can further enhance the manuscript's quality. These include explicitly clarifying whether the proportional odds assumption was tested and expanding on the implementation challenges of proposed policies in the Bangladeshi context. Overall, the manuscript is well-prepared and, with these refinements, is suitable for publication, contributing meaningfully to the global conversation on adolescent mental health.

Author's response: Thank you for your nice comment. Your insightful comment helps us a lot to improve the overall quality of this manuscript. We have addressed this comment in methodology and conclusion section. We tested and shared the findings of the proportional odds assumption in updated manuscript. Besides we have also described the implementation challenges of proposed policies in the Bangladeshi context. Thank you for being optimistic and helping to improve the overall quality of this manuscript.

---

## [Editor Report · Decision Letter 2]

20 Feb 2025

Prevalence and socio-demographic correlates of depression and anxiety among late adolescents (15 to 21 years) in Mymensingh division, Bangladesh: A cross-sectional study

PONE-D-24-28645R2

Dear Dr. Islam,

We’re pleased to inform you that your manuscript has been judged scientifically suitable for publication and will be formally accepted for publication once it meets all outstanding technical requirements.

Kind regards,

AKM Alamgir, PhD

Academic Editor

PLOS ONE
---

## [Editor Report · Acceptance letter]

PONE-D-24-28645R2

PLOS ONE

Dear Dr. Islam,

I'm pleased to inform you that your manuscript has been deemed suitable for publication in PLOS ONE. Congratulations! Your manuscript is now being handed over to our production team.

Kind regards,

on behalf of

Dr AKM Alamgir

Academic Editor

PLOS ONE